# Diagnostic Value of Galectin-3 in Exacerbations of Chronic Obstructive Pulmonary Disease

**DOI:** 10.3390/medicina60040529

**Published:** 2024-03-24

**Authors:** Nurcan Kırıcı Berber, Siahmet Atlı, Ayşegül Altıntop Geçkil, Mehmet Erdem, Tuğba Raika Kıran, Önder Otlu, Erdal İn

**Affiliations:** 1Department of Chest Diseases, Malatya Turgut Özal University, Malatya 44210, Turkey; aysegul.altintop@gmail.com; 2Department of Chest Diseases, Van Training and Research Hospital, Van 65100, Turkey; siahmetatli942@gmail.com; 3Department of Medical Biochemistry, Malatya Turgut Özal University, Malatya 44210, Turkey; mehmet.erdem@ozal.edu.tr (M.E.); raika.kiran@ozal.edu.tr (T.R.K.); onder.otlu@ozal.edu.tr (Ö.O.); 4Department of Pulmonary Diseases, Faculty of Medicine, İzmir University of Economics, İzmir 35330, Turkey; inerda@gmail.com

**Keywords:** chronic obstructive pulmonary disease, acute exacerbation, galectin-3, biomarker, systemic inflammation

## Abstract

*Background and Objectives*: Chronic obstructive pulmonary disease (COPD) is a chronic inflammatory disease characterized by acute exacerbations. Systemic inflammation and oxidative stress play an important role in the pathogenesis of COPD. Exacerbations in COPD reduce the quality of life and are associated with rapid disease progression. Galectin-3 is a beta-galactoside-binding lectin of approximately 30 kDa with pro-inflammatory and pro-fibrotic properties. This study aims to analyze the efficacy of serum galectin-3 in predicting exacerbations in COPD patients. *Materials and Methods*: Baseline demographic and clinical characteristics of all patients were recorded and blood samples were collected. A total of 58 consecutive COPD patients, including 28 patients (19 male and 9 female) with stable COPD and 30 patients (23 male and 7 female) with acute exacerbation of COPD (AECOPD), were included in the study. *Results*: Serum galectin-3 levels were significantly higher in the AECOPD group compared to the stable COPD group. A logistic regression analysis revealed that increased galectin-3 levels and disease duration were independent predictors of COPD exacerbation (OR = 5.322, 95% CI: 1.178–24.052, *p* = 0.03; and OR = 1.297, 95% CI: 1.028–1.635, *p* = 0.028; respectively). *Conclusions*: The results of our study demonstrated that Galectin-3 was a strong and independent predictor of exacerbations in COPD patients.

## 1. Introduction

Chronic Obstructive Pulmonary Disease (COPD), a widespread illness, is defined by continual respiratory symptoms and constrained airflow, ranking among the primary three contributors to global mortality [1]. COPD is characterized by chronic inflammation in the airways, lung parenchyma, and pulmonary vascular structures. In addition to local inflammation in the lungs, systemic inflammation markers are known to increase in COPD, especially in advanced stages and during exacerbations [2]. The presence of cardiovascular comorbidities in COPD patients is associated with acute exacerbations and poor prognosis. Cardiovascular comorbidities can be detected in approximately one third of COPD patients, which are independent of traditional risk factors [3]. Although the relationship between COPD and cardiac comorbidities has not been fully explained, several mechanisms have been identified, including systemic inflammation, oxidative stress, endothelial cell apoptosis, and anti-endothelial cell antibodies that induce endothelial cell dysfunction [4]. 

Within this framework, it was reported that endothelial dysfunction, especially affecting the pulmonary vascular bed, had an important role in the pathogenesis of COPD, and markers indicating endothelial dysfunction were increased in patients with COPD [5]. Moreover, exacerbations in COPD patients aggravate disease-related morbidity and mortality [1]. An early recognition of exacerbations is vital for guiding treatment strategies and the hospitalization of high-risk patients. Therefore, biomarkers that can predict disease prognosis and exacerbations can be useful agents under such conditions.

Galectin-3, a beta-galactoside-binding lectin weighing between 29 and 35 kDa, is recognized for its role in amplifying infection impact through fostering inflammatory reactions in the host and facilitating the secretion of diverse cytokines such as IL-6 and TNF-alpha [6]. It serves as a marker for both inflammation and fibrosis, exhibiting significant expression in myeloid cells (including dendritic cells, neutrophils, and monocytes), as well as in epithelial and endothelial cells, alveolar macrophages, and fibroblasts [7]. Galectin-3 influences the development stages of immune cells, angiogenesis, and the healing reaction subsequent to lung damage. After a lung injury occurs, there is an escalation in galectin-3 expression on the surface of alveolar epithelial cells, likely due to the lung’s re-epithelialization process [7,8].

Therefore, studies showed that in most cases of severe COVID-19, hyperinflammation and lung damage, called a cytokine storm, develops, which is likely to be associated with macrophage activation syndrome, while Galectin-3 is thought to play an important role in regulating the inflammatory response, fibrosis, and infection progression in COVID-19 [7,8]. Other studies also reported that significant amounts of galectin-3 are released from inflammatory cells in patients with severe COVID-19 and that increased galectin-3 serum levels correlate with disease severity [9,10]. Increased levels of galectin-3 in serum have been reported to be associated with worse outcomes and lower survival in patients with acute respiratory distress syndrome (ARDS) [11]. Some other previous studies on galectin-3 have shown that galectin-3 can be used as a diagnostic or prognostic biomarker for heart failure, renal failure, and some types of cancer (thyroid carcinoma, hepatocellular carcinoma (HCC), endometrial tumor, osteosarcoma, etc.) [12].

Although it was discovered that galectin-3 levels were increased locally in the airways of severe COPD patients in the stable cycle and galectin-3 levels were also increased in the blood during exacerbations, its role in COPD patients has not been fully elucidated clinically [13,14].

This study aims to determine serum galectin-3 levels in COPD patients in stable and exacerbation periods, to determine the possible relationship between galectin-3 and various biomarkers, and to analyze the power of galectin-3 in predicting exacerbations.

## 2. Materials and Methods

### 2.1. Study Design and Subjects

A total of 58 consecutive COPD patients, including 28 patients (19 male and 9 female) with stable COPD and 30 patients (23 male and 7 female) with acute exacerbation of COPD (AECOPD), who met the study criteria and who applied to Malatya Turgut Özal University Training and Research Hospital Chest Diseases outpatient clinic between February and May 2022, were included in the study. 

The diagnosis of COPD was confirmed utilizing the Global Initiative for Chronic Obstructive Lung Disease (GOLD) criteria, which necessitated conducting spirometry tests on individuals displaying characteristic symptoms of the disease (such as dyspnea, chronic cough, or sputum production) and/or those exposed to risk factors. Acute exacerbation of COPD (AECOPD) was defined as a sudden deterioration in respiratory symptoms necessitating additional treatment. The demographic data, clinical parameters, and spirometry measurements of the patients were recorded. Blood samples were collected after completion of the medical history, physical examination, and respiratory function tests.

Patients with any organ malignancy, rheumatologic disease, sepsis, respiratory pathology other than COPD (asthma, pneumonia, pulmonary embolism, pulmonary hypertension, interstitial lung disease, etc.), acute myocardial infarction, acute coronary syndrome, heart failure, renal failure, diabetes, and pregnancy were excluded.

### 2.2. Measurement of Serum Galectin-3 Levels

The measurement involved preparing serum samples that were stored at −80 °C while ensuring that the necessary conditions were met. Galectin-3 levels were measured using commercial enzyme-linked immune assay [Enzyme-Linked Immunosorbent Assay (ELISA)] kits from (Cloud-Clone Corp., Cat. No: SEA303Hu), following the manufacturer’s recommendations. The detection range for galectin-3 was 0.156–10 ng/mL.

### 2.3. Statistical Analysis

The statistical analysis was conducted utilizing IBM SPSS Statistics 21 software (Version 25.0; IBM Corp, New York, NY, USA). The results were expressed as mean ± standard deviation (SD). A *p*-value less than 0.05 was considered statistically significant. Student’s *t*-test was employed to compare two independent samples, while the chi-square (X^2^) test was used to compare gender distribution between groups. The Pearson correlation test was utilized to assess parametric values. The cut-off value for galectin-3 was determined using the Receiver Operating Characteristic (ROC) analysis method, and sensitivity and specificity values for galectin-3 were calculated based on this cut-off value. The Area under the Curve (AUC) value was determined using the ROC curve. Binary logistic regression analysis was utilized to identify potential predictors of COPD exacerbation. 

## 3. Results

### 3.1. Comparison of Groups

The study comprised a total of 58 COPD patients, with 28 having stable COPD and 30 experiencing AECOPD. There were no statistically significant differences between the two groups regarding age, gender, and BMI (*p* = 0.07 for age, *p* = 0.45 for gender, X^2^ = 0.563, *p* = 0.34 for BMI).

The mean serum galectin-3 levels for patients with stable COPD and those with AECOPD were 2.08 ± 0.52 and 2.86 ± 0.9 ng/mL, respectively (*p* < 0.001) (Figure 1). 

A comparison of all demographic and laboratory data of COPD patients is presented in detail in Table 1.

### 3.2. ROC Curve Analysis

When the cut-off value for galectin-3 for predicting exacerbations in COPD patients was set at 2.19 ng/mL, sensitivity and specificity were 77% and 58%, respectively (AUC: 0.755, 95% Cl: 0.611–0.866, *p* = 0.001) (Figure 2).

### 3.3. Correlation Analysis

The association between serum galectin-3 levels and various parameters was examined through linear regression models. Within this framework, serum galectin-3 levels were positively correlated with disease duration (Figure 3a) and N-terminal pro-B-type natriuretic peptide (NT-proBNP) levels (r = 0.39, *p* = 0.003; and r = 0.50 *p* < 0.001, respectively) (Figure 3b), whereas serum galectin-3 levels were negatively correlated with partial oxygen pressure (PaO_2_) (Figure 3c) and albumin levels (r = −0.47, *p* < 0.001; and r = −0.27, *p* = 0.04, respectively) (Figure 3d). There was no statistically significant correlation between serum galectin-3 and neutrophil count (Figure 3e), and CRP levels (Figure 3f).

### 3.4. Logistic Regression Analysis

Binary logistic regression analysis was performed to identify potential predictors of exacerbations in COPD patients. In the multivariable logistic regression model, galectin-3 and disease duration were found to be independent predictors of exacerbations in COPD patients (OR = 5.322, 95% CI: 1.178–24.052, *p* = 0.03; and OR = 1.297, 95% CI: 1.028–1.635, *p* = 0.028, respectively).

### 3.5. Summary of the Results of the Study

We present the summary of the results of the study in Figure 4.

## 4. Discussion

In our study, in which we analyzed the efficacy of galectin-3 in predicting exacerbations and disease severity in COPD patients, we observed that galectin-3 levels were significantly higher in AECOPD patients than in stable COPD patients. Furthermore, galectin-3 was found to be an independent predictor of exacerbation, and correlated positively with disease duration and NT-proBNP and negatively with PaO_2_ and albumin. Exacerbations in COPD significantly impact the health status and progression of the disease, often necessitating hospitalizations [1]. COPD is marked by systemic inflammation alongside pulmonary vascular remodeling, coupled with aberrant inflammatory reactions within the lungs. Exacerbations are known to exacerbate these inflammatory processes [15,16]. The literature indicates that the levels of numerous inflammatory biomarkers rise during the onset of exacerbation in COPD patients, with subsequent reductions observed during the treatment phase [17]. Additionally, it is known that the levels of traditional inflammation markers such as CRP, fibrinogen, and leukocyte count are elevated during exacerbation in COPD patients [18,19,20]. However, studies on the value of galectin-3 COPD exacerbation are limited. Two studies demonstrated that galectin-3 levels were increased in blood and lung tissues in COPD patients [13,14]. In the research conducted by Pilette et al., it was observed that galectin-3 immunostaining exhibited an elevation in the epithelium of the small airways among COPD patients compared to both non-smoker and smoker controls. Moreover, the study revealed a correlation between galectin-3 levels in lung tissues and heightened epithelial damage and airflow obstruction in individuals with COPD [13]. In another study, covering 44 COPD patients, it was discovered that galectin-3 levels were higher during the exacerbation period compared with the convalescence phase. Moreover, in the same study, galectin-3 was discovered to be associated with systemic inflammation and smoking in COPD patients. Accordingly, a positive correlation was observed between galectin-3 levels and CRP and NT-proBNP levels in AECOPD patients [14]. Similarly, in our study, serum galectin-3 levels were significantly higher in AECOPD patients compared to stable COPD patients, while galectin-3 was discovered to be an independent predictor of exacerbations in the multivariable logistic regression analysis. Within this framework, it is known that the levels of systemic inflammatory markers increase with exacerbations in COPD patients and there is a relationship between increased biomarker levels and disease severity [17,21]. 

Galectin-3 is a pro-inflammatory marker and its levels may increase with the increased inflammatory response in the early period of exacerbations. In our study, when the cut-off value for galectin-3 was set to 2.19 ng/mL, it was determined to predict exacerbations with 77% sensitivity and 58% specificity. Accordingly, our findings suggest that galectin-3 is a useful marker for the early detection of exacerbations. In the initial scientific investigation assessing the prognostic significance of Galectin-3 in acute respiratory failure induced by COVID-19 disease, it emerged as a predictor of mortality, admission to the intensive care unit, and the development of severe ARDS in COVID-19 patients [8]. Another study examining Galectin-3 in COVID-19 patients identified in-hospital mortality and/or the requirement for invasive mechanical ventilation in severe COVID-19 cases as severe outcomes. In this study, Galectin-3 levels were notably elevated in patients experiencing severe outcomes compared to those without [6]. According to the existing literature, cardiac biomarker levels tend to rise in COPD patients, particularly during exacerbation episodes, and cardiac impairment is linked to a poorer prognosis [22,23]. 

NT-proBNP, released by cardiomyocytes in response to ventricular strain, serves as an indicator of right ventricular dysfunction [24]. Pulmonary hypertension frequently arises as a complication of COPD, and its presence correlates with reduced survival rates and an increased frequency of exacerbations [25]. Although endothelial dysfunction in the pulmonary vascular bed in patients with COPD was observed in the early stages of the disease, it is also known to increase significantly with the severity of the disease and during periods of exacerbation [26]. Increased inflammation may trigger galectin-3 release from various host cells [24]. Galectin-3 is known to be expressed in both alveolar epithelial and endothelial cells [15]. Hence, elevated galectin-3 levels during exacerbations may be indicative of both parenchymal and vascular damage. In line with this observation, our study revealed elevated NT-proBNP levels in AECOPD patients, with a significant positive correlation between galectin-3 and NT-proBNP levels. Increased galectin-3 levels may be associated with increased pulmonary vascular resistance due to hypoxia and oxidative stress in addition to systemic and pulmonary inflammation. Indeed, in our study, a negative correlation was found between serum galectin-3 levels and PaO_2_ levels. Albumin, characterized as a negative acute-phase protein, possesses antioxidant properties. During the acute-phase response, circulating concentrations of albumin typically decrease [27]. 

In the literature, it was reported that serum albumin levels decreased in COPD as a result of malnutrition and increased inflammatory response. Furthermore, low albumin concentration was reported to be a risk factor for acute respiratory failure and mortality in patients with COPD [28]. 

To corroborate these findings, we observed significantly lower serum albumin levels in the AECOPD group. Furthermore, our study revealed a negative correlation between galectin-3 levels and albumin levels. Accordingly, increased systemic inflammatory response and oxidative stress seem to increase galectin-3 concentrations and decrease albumin levels in COPD patients. Previous studies showed that low albumin levels are a predictor of severe morbidity and mortality in COVID-19 patients [29,30,31].

A possible mechanism of hypoalbuminemia in COVID-19 patients is albumin leakage into the interstitial space due to increased capillary membrane permeability as a result of a cytokine storm, especially in patients with severe COVID-19 disease; albumin production was determined to be due to liver damage.

The limitations of our study include the fact that our study was a single-center study. The lack of patients with ARDS/critically severe pneumonia, adequate intensive care, and mechanical ventilation among our study patients is a limitation of our study. Another limitation is that a long-term follow-up of the patients was not performed. 

## 5. Conclusions

In conclusion, this study demonstrated the importance of galectin-3 as an inflammatory biomarker associated with hypoxia and systemic inflammation in patients with COPD. The results of our study demonstrated that elevated galectin-3 levels have potential value as a robust and independent marker of exacerbation in COPD. However, prospectively designed and experimental studies are required to fully elucidate the effects of galectin-3 on the pathogenesis and outcomes of COPD.

## Figures and Tables

**Figure 1 medicina-60-00529-f001:**
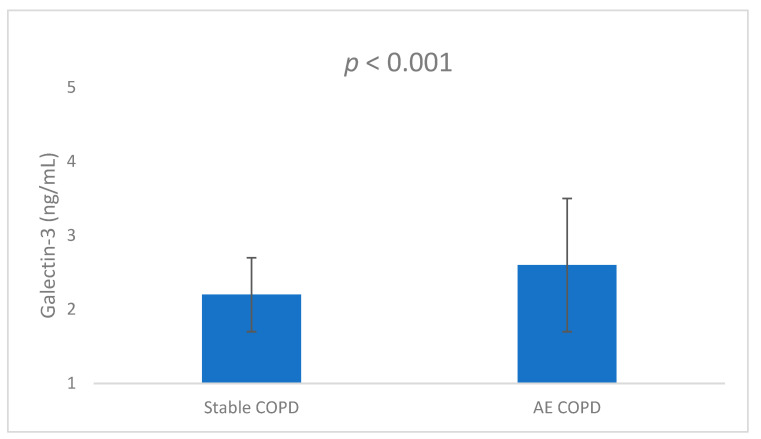
Comparison of the serum galectin-3 levels of stable COPD and AECOPD groups.

**Figure 2 medicina-60-00529-f002:**
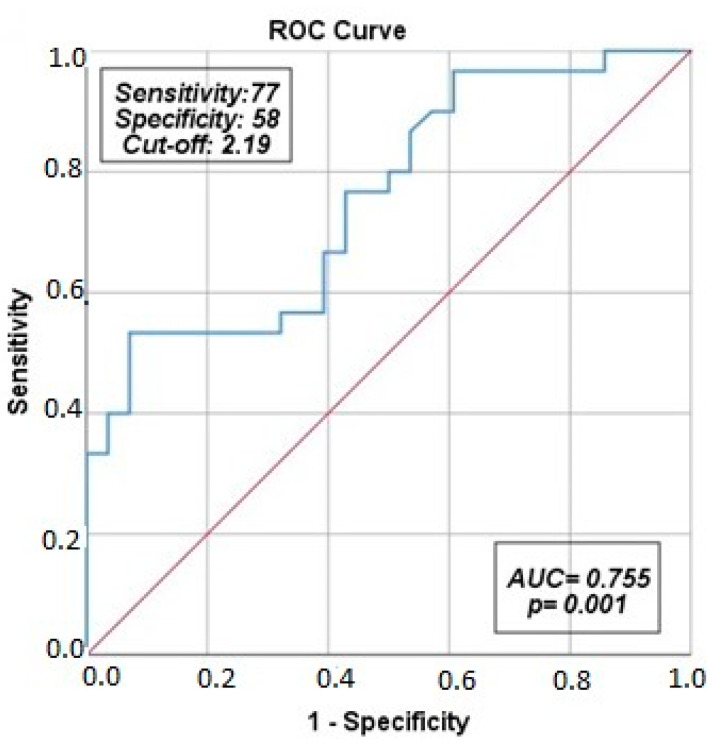
ROC curve analysis of the utility of galectin-3 to AECOPD. The red line is the reference and the blue line is the ROC curve.

**Figure 3 medicina-60-00529-f003:**
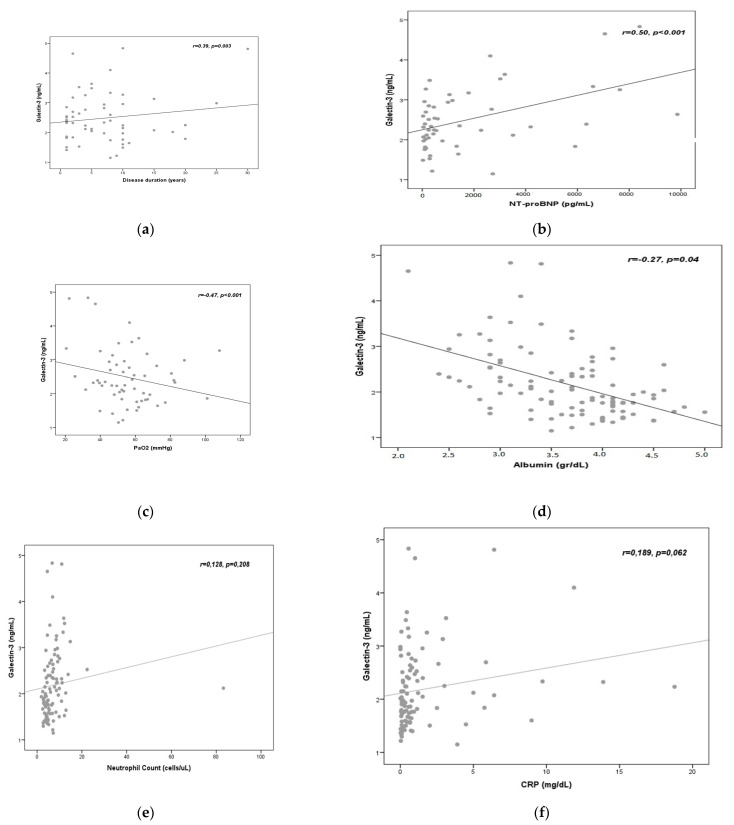
Correlation analysis of the serum galectin-3 levels and disease duration (years) (**a**). Correlation analysis of the serum galectin-3 and N-terminal pro-B-type natriuretic peptide (NT-proBNP) levels (**b**). Correlation analysis of the serum galectin-3 levels and partial oxygen pressure (PaO_2_) values (**c**). Correlation analysis of the serum galectin-3 and albumin levels (**d**). Correlation analysis of the serum galectin-3 and neutrophil count levels (**e**). Correlation analysis of the serum galectin-3 and CRP levels (**f**).

**Figure 4 medicina-60-00529-f004:**
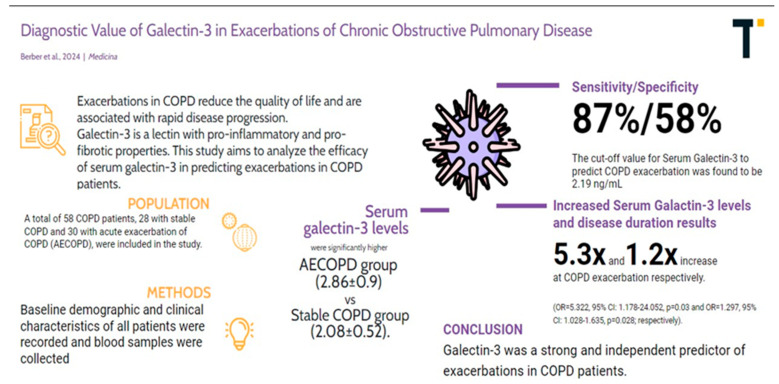
Figure showing the summary of the article.

**Table 1 medicina-60-00529-t001:** Comparison of the demographical and laboratory data of two groups of patients with COPD.

Variable	COPD Group	Mean	Std. Deviation	*p*
Age (years)	Stable COPD	66.14	10.03	0.07
	COPD with AE	71.57	12.36	
BMI (kg/m^2^)	Stable COPD	27.75	7.78	0.34
	COPD with AE	26.15	4.12	
Disease duration (years)	Stable COPD	5.46	3.24	0.001
	COPD with AE	9.47	4.11	
Smoking (pack-years)	Stable COPD	55.11	16.69	0.13
	COPD with AE	46.38	15.72	
Leukocyte (10^3^/L)	Stable COPD	10.39	3.95	0.78
	COPD with AE	10.13	3.12	
Hemoglobin (g/dL)	Stable COPD	14.16	2.61	0.99
	COPD with AE	14.15	2.51	
Thrombocyte (10^3^/L)	Stable COPD	288.50	65.71	0.09
	COPD with AE	246.77	49.04	
Creatinine (mg/dL)	Stable COPD	0.88	0.20	0.98
	COPD with AE	0.88	0.24	
ALT (U/L)	Stable COPD	30.46	4.36	0.98
	COPD with AE	30.26	4.45	
AST (U/L)	Stable COPD	22.11	13.79	0.66
	COPD with AE	23.80	15.77	
Albumin (g/dL)	Stable COPD	3.46	0.48	0.009
	COPD with AE	3.13	0.45	
CRP (mg/L)	Stable COPD	1.20	1.11	0.01
	COPD with AE	4.53	3.44	
NT-proBNP (pg/mL)	Stable COPD	606.24	399.18	0.001
	COPD with AE	2187.56	1159.17	
FVC (%)	Stable COPD	53.78	25.41	0.03
	COPD with AE	41.50	17.53	
FEV1 (%)	Stable COPD	44.93	22	0.03
	COPD with AE	33.60	17.41	
FEV1/FVC	Stable COPD	63.82	6.96	0.49
	COPD with AE	62.47	8.10	
pH	Stable COPD	7.37	0.06	0.53
	COPD with AE	7.36	0.06	
PaO_2_ (mmHg)	Stable COPD	58.27	9.35	0.008
	COPD with AE	50.9	11.18	
SaO_2_ (%)	Stable COPD	87.01	6.04	0.01
	COPD with AE	80.49	11.72	
PaCO_2_ (mmHg)	Stable COPD	46.32	15.31	0.03
	COPD with AE	55.46	17	
Galectin (ng/mL)	Stable COPD	2.08	0.52	0.001
	COPD with AE	2.86	0.90	

The data are presented as mean ± standard deviation (SD). Student’s *t*-test was employed to compare two independent samples, while the chi-square (X^2^) test was utilized to compare gender distribution between groups. AE: Acute exacerbation; SaO_2_: Oxygen saturation; ALT: Alanine transaminase; AST: Aspartate transaminase; NT-proBNP: NT-proBNP: N-terminal pro-brain natriuretic peptide; CRP: C-Reactive protein; BMI: Body mass index; N-terminal pro-brain natriuretic peptide; FVC: Forced vital capacity; FEV1: Forced expiratory volume in 1 s; pH: Potential hydrogen; PaCO_2_: Partial carbon dioxide pressure; PaO_2_: Partial oxygen pressure; SaO_2_: Oxygen saturation.

## Data Availability

The datasets are not publicly available but are available from the corresponding author upon reasonable request.

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
