# Peer review of "Diagnostic Value of Galectin-3 in Exacerbations of Chronic Obstructive Pulmonary Disease"

_medicina, 2024, doi:10.3390/medicina60040529_

Round 1

Reviewer 1 Report

Comments and Suggestions for Authors

This is an interesting contribution suggesting increased Galectin-3 in serum levels correlates with exacerbation of COPD. The following minor issues may be considered:

1.       Reduce the length of the abstract and add a graphical abstract for a quick grasp of the message.

2.       Figure 1: show single data plots and for the y-axis remove the zero: 1-5 ng/ml and add the standard curve for the data.

3.       Figure 3: add the correlation of Galectin-3 with CRP, what about neutrophils?

Add List of abbreviations

. The following minor issues may be considered:

1.       Reduce the length of the abstract and add a graphical abstract for a quick grasp of the message.

2.       Figure 1: show single data plots and for the y-axis remove the zero: 1-5 ng/ml and add the standard curve for the data.

3.       Figure 3: add the correlation of Galectin-3 with CRP, what about neutrophils?

     4. Add List of abbreviations

Author Response

Medicina] Manuscript ID: medicina-2904854 - Minor Revisions

First of all, we would like to thank our reviewers for their valuable contributions to our study. We have made the following adjustments in accordance with the recommendations of the reviewers.

Q1-Reduce the length of the abstract and add a graphical abstract for a quick grasp of the message.

A1- We reduced the length of the abstract and added a graphical abstract for a quick grasp of the message. Graphical abstract has been added as supplementary document due to document template has no Graphical abstract area

Q2 Figure 1: show single data plots and for the y-axis remove the zero: 1-5 ng/ml and add the standard curve for the data.

A2. We received consultancy from a biostatistician for the Figure 1 and changed the design of graphic which is more suitable for our data in line with his suggestions

Q3. Figure 3: add the correlation of Galectin-3 with CRP, what about neutrophils?

A3. It was requested to include the graph of the correlation between Galectin-3 and CRP and neutrophils, but since the stated correlations were not statistically significant (for CRP r=0.189, p=0.062 and for neutrophil count r=0.128, p=0.208). We preferred not to include it as a graph in the article. Instead, it was stated as a short text at the relevant place in the article.

Q4. Add List of abbreviations

A4. To accommodate the reviewer's suggestion, we re-examined the journal's Instructions for Authors area. Instructions dedicates that “Abbreviations should be defined the first time they appear in each of three sections: the abstract; the main text; the first figure or table. When defined for the first time, the abbreviation should be added in parentheses after the written-out form.” Also journal manuscript template provided by Medicina has no area for list of abbreviations. Authors defined the Abbreviations in each sections as mentioned in instructions instead of adding a List of abbreviations

 And

 -Institutional email address:

1.Address for  Dr. Siahmet Atlı;  siahmet.atli@saglık.gov

2.Address  for  Dr. Ayşegül Altıntop Geçkil ;  aysegul.geckil@ozal.edu.tr

  1. Address for Prof. Dr.Erdal İn;  erdal.in@mhp.com.tr

Sincerely,

Nurcan Kırıcı Berber

Department of Chest Diseases, Malatya Turgut Ozal University Faculty of Medicine, Malatya, Turkey

nurcan.berber@ozal.edu.tr

+90-422-211-8019

Reviewer 2 Report

Comments and Suggestions for Authors

Nurcan Kırıcı Berber et al., determine serum galectin-3 levels in COPD patients in stable and exacerbation periods to determine the possible relationship between galectin-3 and Exacerbations of Chronic Obstructive Pulmonary Disease.

Galectin-3 has been widely used in several pathologies; the manuscript is interesting and could serve as a reference in the future. However, the authors mention that COPD is one of the diseases with the highest mortality, and including only 58 samples seems too few to use as significant.

In addition, some minor issues are recommended to the authors

Delete lines 167-169, as they appear to be recommendations.

Add study limitations

Add more references

Lack of discussion of your findings
